# Crystal Packing Differences as a Key Factor for Stabilization of the N-Terminal Fragment of the Human HINT1 Protein

**Rafał Dolot** *, **Aleksandra Mikołajczyk** and **Barbara Nawrot**

Centre of Molecular and Macromolecular Studies, Polish Academy of Sciences, Sienkiewicza 112, 90-363 Lodz, Poland; aleksandraseda@gmail.com (A.M.); barbara.nawrot@cbmm.lodz.pl (B.N.)
* Correspondence: rafal.dolot@cbmm.lodz.pl; Tel.: +48-42-6803325

**Abstract:** Histidine triad nucleotide-binding protein 1 (HINT1) is the oldest and most widely distributed branch of the histidine triad superfamily of proteins. The HINT1 protein plays an important role in various biological processes and has been found in many species. Here we report the first nearly complete structure of the human HINT1 protein at 1.43 Å resolution obtained from a crystal of the $P2_12_12_1$ orthorhombic space group. The final structure has an $R_{cryst}$ = 22.4% ($R_{free}$ = 27.7%) and contains a fragment of the N-terminal part that was not determined in the previously deposited structures. In addition, selective binding of the L-malate ion was detected, which had not been observed previously.

**Keywords:** histidine triad; HINT; protein crystallography; crystal packing; ligand binding

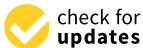



## 1. Introduction

Histidine triad nucleotide-binding protein 1 (HINT1) belongs to a branch of the histidine triad protein superfamily (HIT), which consists mainly of mono- and dinucleotide hydrolases and nucleotide transferases. They have a characteristic active site motif HXHX-HXX, where X is a hydrophobic residue [1]. HINT1 and other HINT proteins are the most highly conserved members of the HIT superfamily, and their homologs are found in many organismal kingdoms, including metazoans [2–4], plants [5], fungi [6], and bacteria [7,8]. HINT1 is expressed in numerous tissues and is present in cell nuclei and cytoplasm, but its function is only partially elucidated. It is involved in transcriptional regulation [9–15] and cell cycle control [16]. In mammals, HINT1 modulates apoptosis in cancer cells and is a potential tumor suppressor [12,14,17–23]. Recent studies have shown that HINT1 plays an important role as a platform protein in the formation of the G protein-coupled receptors (GPCRs) MOR (μ-opioid receptor) and CNR1 (cannabinoid receptor type 1) [24,25].

The first crystal structure of HINT1 was published in 1996 [26], and the protein was first described as an inhibitor of protein kinase C [27] and later as PKCI (protein kinase C-interacting) [28]. The name "HINT" resulted from structural analysis [29]. Currently, many HINT1 structures are deposited in the PDB, including human [26,30–32], rabbit [29,33,34], and bacterial homologs [8–10]. The deposited structures show HINT either in the apo state [26] or as complexes with selected ligands—either substrate analogs or products of their hydrolysis [8,30,33]. The best resolution of data for HINT1 to date was 0.95 Å for a human homolog (PDB code 6j64) [35].

In vitro studies indicate that human HINT1 binds multiple nucleotides, e.g., adenosine-5′-monophosphate (AMP), adenosine-5′-diphosphate (ADP), and the diadenosine polyphosphates 5′,5″-diadenosine triphosphate (Ap3A) and 5′,5″-diadenosine tetraphosphate (Ap4A) [30]. Rabbit HINT1 (rHINT1) also binds certain purine nucleosides and nucleotides [29]. HINT proteins exert phosphoramidase activity toward adenosine 5′-*O*-phosphoramidate (AMP-NH2) [2]. Lysyl adenylate (produced by lysyl-tRNA synthetase

(LysRS)) is also considered a substrate for HINT1 hydrolase [36,37]. HINT1-assisted hydrolysis of P–N bonds in nucleoside 5'-*O*-phosphoramidates [38] and desulfuration of nucleoside 5'-*O*-phosphorothioates [35,39] have been documented.

To date, all mammalian HINT1 structures deposited in the Protein Data Bank lack the first eleven N-terminal amino acids, and electron density maps show only the remaining 115 residues. The length of this truncated structure is similar to that of *E. coli* HINT1, which is shorter than the mammalian protein (119 amino acids). In the present manuscript, we show and discuss data demonstrating a nearly complete structure of the N-terminal part of the protein chain, refined to 1.43 Å. This improvement may be related to the modified arrangement of the HINT1 molecules in an asymmetric unit (compared to the previously reported crystal structures obtained in the same space group) when modified crystallization conditions were used.

## 2. Materials and Methods

### 2.1. Cloning, Expression, and Purification of hHINT1

Human HINT1 protein was expressed and purified in an *E. coli* BL21 strain using a pSGA02-hHINT1 plasmid as previously described for rabbit HINT1 [40]. Purification was performed in one step by AMP-agarose (Sigma) affinity chromatography. The homogeneous protein preparation was dialyzed against a buffer containing 20 mM Tris-HCl (pH 7.5) and 150 mM NaCl, concentrated to a protein concentration of 10 mg mL$^{-1}$, frozen in liquid nitrogen, and stored at −80 °C.

### 2.2. Crystallization of hHINT1

Human HINT1 was crystallized by a standard hanging drop variant of the vapor diffusion method using a protein solution with a concentration of 6 mg mL$^{-1}$. Suitable crystals were obtained during crystallization experiments under Jena Biosciences JBScreen PACT ++ conditions. Aliquots of a 1.5 μL protein solution were mixed with a 1.5 μL precipitation solution and the crystallization droplets were suspended over a 500 μL precipitation solution. After 72 h of storage at 281 K under PACT ++ 2 D4 conditions (25% *w/v* PEG 1500 and 0.1 M MMT buffer pH 7.0), several suitable crystals (plate-shaped with typical dimensions 0.6 × 0.35 × 0.05 mm, Figure 1) were obtained.

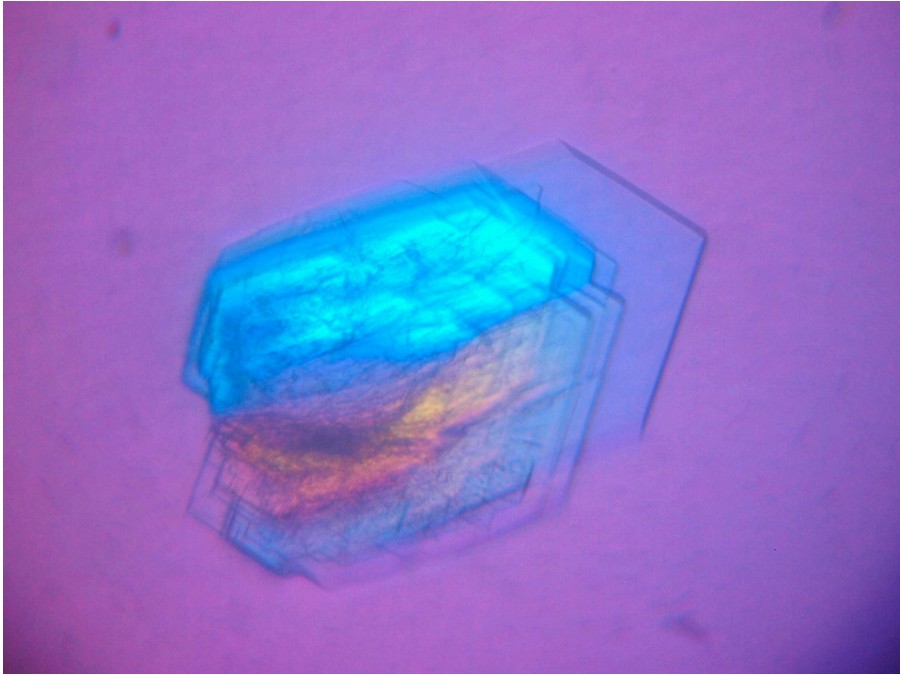

**Figure 1.** Crystals of hHINT1 obtained from JBScreen PACT++2 D4 conditions.

## *2.3. Data Collection, Structure Determination, and Refinement*

The crystals studied did not require cryoprotection, but were deposited on a very thin film of crystallization buffer. The excess liquid was removed by gently touching the mounting loop on the plate surface, and the crystal was then cooled directly in an $N_2$ stream. Diffraction data were acquired with synchrotron radiation using a Dectris Pilatus 6M detector at the BL14.1 beamline of the BESSY II synchrotron (Berlin, Germany) [41]. The data were processed, integrated, and scaled using an XDS package [42]. Data acquisition and processing statistics are listed in Table 1.

**Table 1.** Crystallographic parameters, data collection and refinement statistics for the HINT1 structure described here. Values in parentheses correspond to the highest resolution shell.

| PDB Entry | 6g9z |
|---|---|
| Space group | $P2_12_12_1$ |
| Wavelength (Å) | 0.96690 |
| Unit-cell parameters | |
| *a* (Å) | 46.08 |
| *b* (Å) | 63.38 |
| *c* (Å) | 76.84 |
| Total no. of reflections | 154,268 |
| Unique reflections | 40,954 |
| Resolution (Å) | 39.52–1.43 (1.51–1.43) |
| Completeness (%) | 96.1 (92.7) |
| *CC*(1/2) (%) | 99.4 (83.3) |
| $R_{merge}$ [a] (%) | 13.7 (70.3) |
| Redundancy | 3.77 |
| Mosaicity (°) | 0.21 |
| Wilson *B* factor (Å$^2$) | 15.0 |
| Mean $I/\sigma(I)$ | 8.78 (1.96) |
| Refinement statistics | |
| No. reflections used in refinement | 38,759 |
| No. reflections used to $R_{free}$ | 2041 |
| $R_{cryst}$ ($R_{free}$) [b] | 0.224/0.277 |
| No. non-H atoms | |
| Protein | 1998 |
| Solvent | 401 |
| Ligands | 9 (malate ion) |
| R.m.s.d. from ideal values | |
| Bond lengths (Å) | 0.016 |
| Bond angles (°) | 1.812 |
| Ramachandran plot [c] | |
| Favored (%) | 98.4 |
| Allowed (%) | 1.2 |
| Outliers (%) | 0.4 |
| Mean B values [d] | |
| Protein (Å$^2$) | 8.53 |
| Malate ion (Å$^2$) | 14.26 |
| Water (Å$^2$) | 25.53 |

[a] $R_{merge} = \sum_h \sum_i |I_i(h) - \langle I(h) \rangle| / \sum_h \sum_i I_i(h)$, where $I_i(h)$ is the intensity of an individual measurement of the reflection and $\langle I(h) \rangle$ is the mean intensity of the reflection. [b] $R_{cryst} = \sum_h ||F_o| - |F_c|| / \sum_h |F_o|$ for all reflections, where $F_O$ and $F_C$ are observed and calculated structure factors, respectively. $R_{free}$ is calculated analogously for the test reflections, randomly selected, and excluded from the refinement. [c] Calculated using *MOLPROBITY*. [d] Calculated using *BAVERAGE*.

The structure was solved by molecular replacement with MOLREP [43], using the protein model of the hHINT1 crystal structure (PDB entry 3tw2 [32]) as a search model. Refinement was performed using REFMAC5 [44]. After calculating an initial electron density map, the structure was completed by alternating manual building cycles (in the program COOT [45]), including rebuilding the main and side chains, adding alternative

conformations of selected residues, loop fragments, ligand, and solvent molecules. All refinement steps were monitored using $R_{cryst}$ and $R_{free}$ values. The stereochemical quality of the resulting model was evaluated using the program MOLPROBITY [46] and the validation tools implemented in COOT. The values of the mean temperature factors for the main and side chains of the protein, the ligands, and the water molecules were calculated using the BAVERAGE program from the CCP4 suite. Interface analysis was performed using an EBI PISA server [47]. The refinement statistics of the described structures are listed in Table 1. All figures were generated using PyMOL v.1.7 [48].

### 2.4. Mass Spectrometry Analysis

Positive ion MALDI-TOF mass spectra were acquired using a Voyager-Elite (PerSeptive Biosystems Inc., Framingham, MA, USA) instrument equipped with a nitrogen laser (337 nm) and operating in linear mode at 20 kV accelerating voltage and delayed extraction. A mixture of 10 mg mL$^{-1}$ solution of trans-3,5-dimethoxy-4-hydroxycinnamic acid with 50% acetonitrile in water was used as the matrix. A 1 μL aliquot of the 10 pmol μL$^{-1}$ protein solution in deionized water was mixed with 1 μL of the matrix on a MALDI plate and air dried. A mass spectrum from at least 100 laser shots was accumulated and analyzed using a Data Explorer ver. 4 (Applied Biosystems, Foster City, CA, USA).

## 3. Results and Discussion

### 3.1. Overall Structure of a New Crystal Form of hHINT1 in Space Group $P2_12_12_1$

The crystal structure of hHINT1 (an extended model refined to 1.43 Å and referred to here as form II, in contrast to previously published form I [26,30,49]) contains two protein molecules as well as 401 water molecules in the asymmetric unit, giving an *R*-factor of 0.224 ($R_{free}$ = 0.277). In addition, an L-malate ion (2-hydroxybutanedioic acid residue), a component of the crystallization buffer, was identified in the nucleotide-binding cleft of one of the chains. Almost all protein residues were in favored/allowed Ramachandran conformations, except Val10 of chain B. A pair of molecules around the non-crystallographic twofold symmetry axis parallel to the *x*-axis of the crystal formed a homodimer (Figure 2). Subunit dimerization was identical to that observed previously and occurs over a broad interfacial region providing multiple polar interactions and salt bridges between chains A and B on the accessible 1902.6 Å$^2$ surface area. Each subunit exhibited the typical HINT1 fold and consisted of five β-strands and two α-helices. In contrast to previously published mammalian HINT1 structures, including those refined to atomic resolution, the detailed electron density analysis in maps $2F_O - F_C$ and $F_O - F_C$ allowed for an extension of the existing model so that the final HINT1 model of the form II included residues 6–126 of the A chain and 9–126 of the B chain. The data obtained allowed for modeling of alternative side-chain rotamers for nine residues of the A chain and ten residues of the B chain, with possible double or multiple conformations.

### 3.2. Two Crystal Forms—The Same Space Group

The form II of HINT1 was crystallized in an orthorhombic $P2_12_12_1$ space group and showed significant differences compared to form I (PDB entry 5klz) [49]. The *b* and *c* unit cell parameters of the II form were smaller by 17% and 4%, respectively (form II: *a* = 46.08 Å, *b* = 63.38 Å, *c* = 76.84 Å; form I: *a* = 45.83 Å, *b* = 75.92 Å, *c* = 80.79 Å). A comparison of the calculated Matthews coefficients for form I (2.49 Å$^3$ Da$^{-1}$; 50.66% solvent content) and form II (2.03 Å$^3$ Da$^{-1}$; 39.51% solvent content) revealed that the crystals of form II were much more densely packed, with a loss of solvent content of more than 10% (Figure 3). The denser crystal packing dramatically increased the number of crystal contacts, from only three detected hydrogen bonds between symmetry-related molecules in form I to thirteen hydrogen bonds and eight salt bridges in form II. The size of the interface with symmetry-related protein chains increased from 940.3 Å$^2$ (form I) to 1673.1 Å$^2$ (form II) [47]. The observed degree of packing for form II was comparable to that previously observed

in crystals of the *C*2 monoclinic space group ($V_M$ = 2.08 Å$^3$ Da$^{-1}$) [32]. Comparisons with other crystal forms listed in the Protein Data Bank are shown in Table 2.

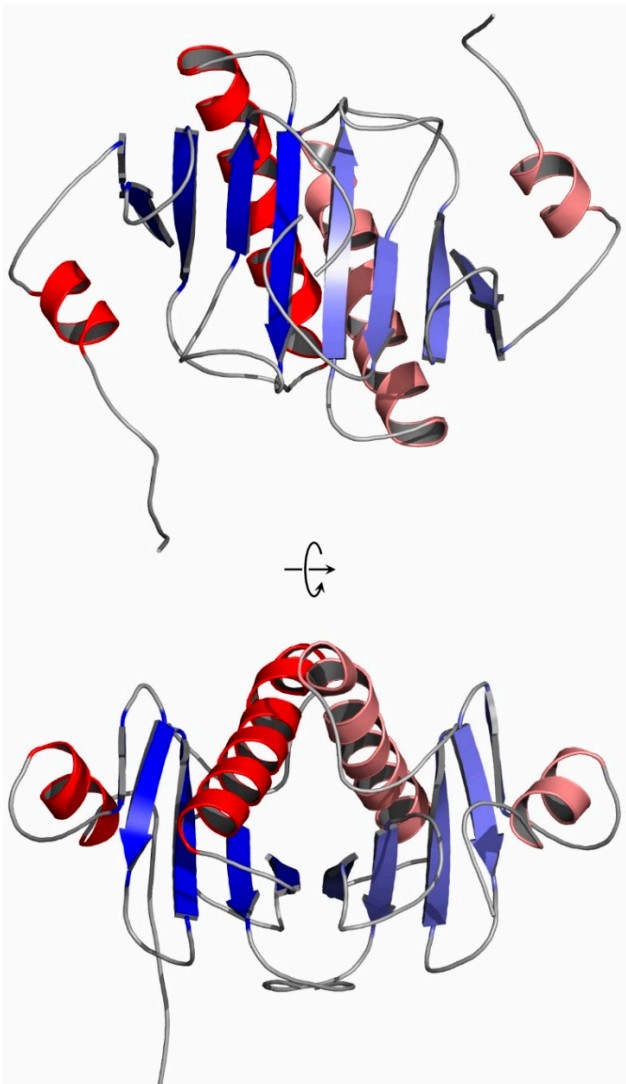

**Figure 2.** Cartoon representation of the hHINT1 dimer with extended N-terminus (based on PDB entry 6g9z). Structural elements are shown in red (α-helices) and blue (β-strands). Each monomer is shown in a darker (chain A) and lighter (chain B) color scheme. The secondary structure elements were assigned using DSSP [50]. The views after a 90° rotation are shown below.

### 3.3. Structure of the N-Terminus

Since the C-terminus of the HINT1 protein does not interact with other proteins, especially transcription factors, it is suggested that the N-terminus or some of its structural motifs may be responsible for the contacts with the binding partners of HINT1 [14]. Among the deposited crystal structures of mammalian HINT1, only the structure described here (PDB entry 6g9z) shows a fragment of the N-terminal part of the polypeptide chain, which has not been shown before. In the previously published structures of HINT1, only residues 12–126 were visible. The ordered state observed here is likely an artefact of crystal packing and is therefore probably not physiologically relevant. In solution, the N-terminal region likely remains disordered unless there are stabilizing protein–protein interactions. To be sure that the studied crystals contained the entire protein chain, mass spectrometry analysis was performed, which confirmed the presence of the intact HINT1 molecule (Figure 4). The major mass peak (*m*/*z* 13,679.8) corresponds to the entire protein chain with the initiation Met residue removed (13,670.7 Da).

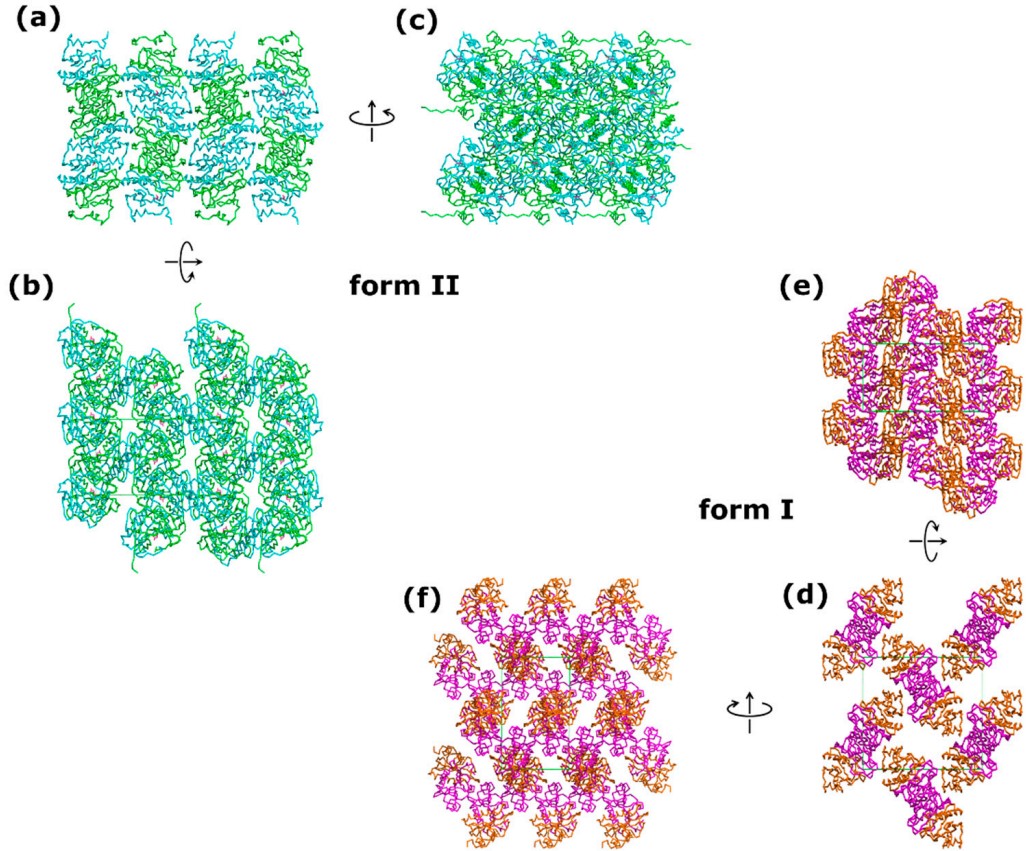

**Figure 3.** Crystal packing diagram of two forms of human HINT1 crystallized in space group $P2_12_12_1$: (**a–c**) hHINT1 form II with visible N-terminus (based on PDB entry 6g9z) and (**d–f**) hHINT1 form I (based on PDB entry 5klz) [49]. Shown are views along crystallographic axes: *a* (**a,d**), *b* (**b,e**), and *c* (**c,f**).

**Table 2.** Examples of different crystal forms of human HINT1 protein presented in Protein Data Bank.

| Space Group | PDB ID | Cell Dimensions a, b, c [Å], $\alpha$, $\beta$, $\gamma$ [°] | Matthews Coefficient [Å$^3$ Da$^{-1}$] | Solvent Content [%] |
|---|---|---|---|---|
| | | Triclinic | | |
| $P1$ | 6j5z | 45.42 45.44 63.92 86.57 86.48 61.35 | 2.07 | 40.54 |
| | | Monoclinic | | |
| $P2_1$ | 4zkl | 63.71 46.42 103.43 90 97.41 90 | 3.03 | 59.45 |
| $P2_1$ | 4zkv | 46.21 79.00 63.88 90 90.10 90 | 2.27 | 45.92 |
| $P2_1$ | 5km0 | 64.33 89.59 46.37 90 90.03 90 | 2.37 | 48.10 |
| $C2$ | 3tw2 | 77.42 46.45 64.03 90 94.42 90 | 2.08 | 40.75 |
| | | Orthorhombic | | |
| $P2_12_12_1$ | 5klz | 45.83 75.92 80.79 90 90 90 | 2.49 | 50.66 |
| | | Tetragonal | | |
| $P4_1$ | 7q2u | 112.89 112.89 43.63 90 90 90 | 2.52 | 51.18 |
| $P4_32_12$ | 6b42 | 39.74 39.74 141.13 90 90 90 | 1.98 | 37.76 |

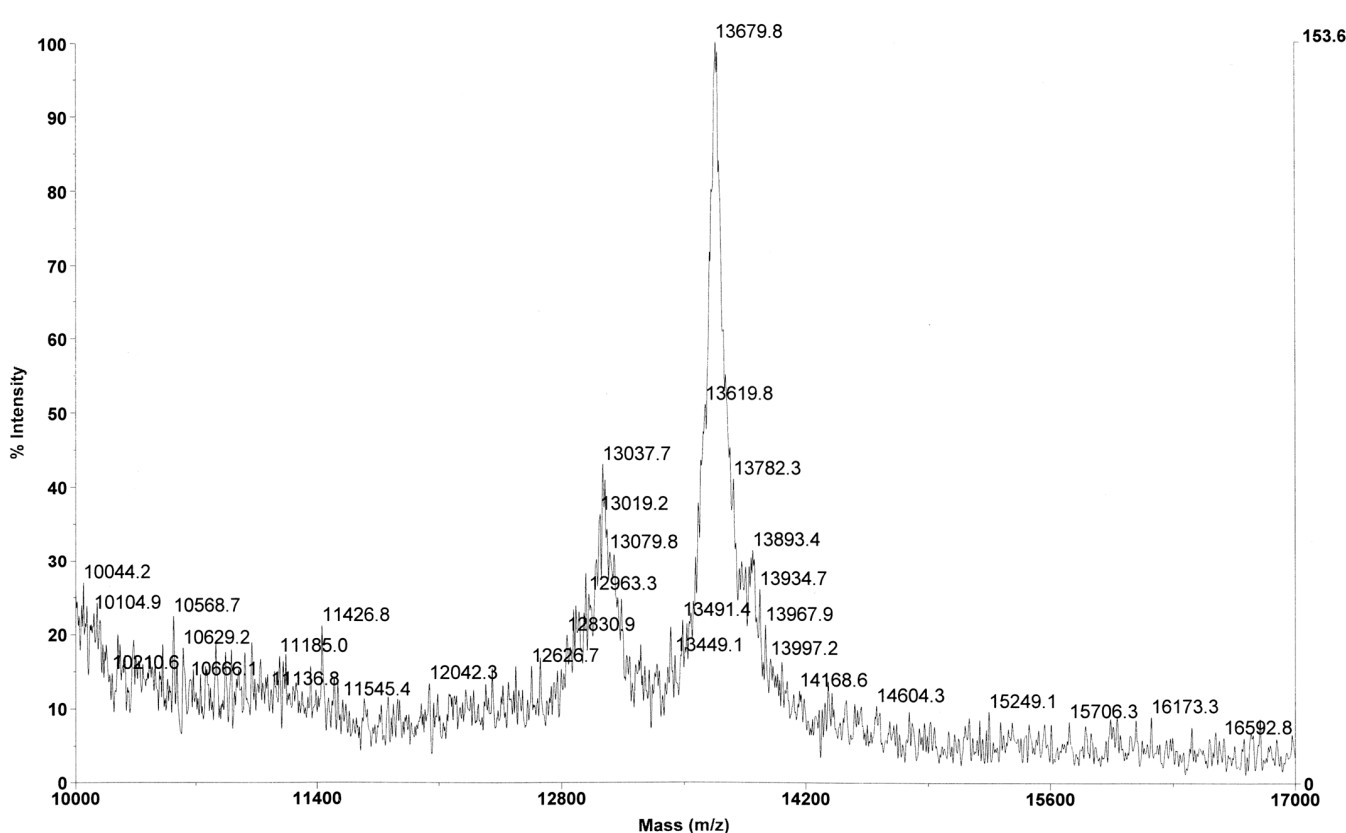

**Figure 4.** Positive-ion spectrum MALDI-TOF, obtained for the sample of hHINT1 protein crystals used in this study.

Detailed analyses of the calculated electron density maps allowed for stepwise modeling of the remaining N-terminal fragments of chains A and B from residues Ala6 (Figure 5a) and Gln9, respectively. Residues 1–5 of chain A and 1–8 of chain B are not visible on the maps, perhaps because of the high flexibility of these regions indicated by high values of the thermal vibration factors of the visible fragments of the N-terminal parts. The different crystal packing of form II led to a decrease in the flexibility of the N-terminus of chain A when interacting with the following symmetry-related molecules: the chain A of SYM1 (4_445) and the chain B of SYM2 (3_545). The interactions arise from hydrogen bonds formed by the carbonyl oxygen of Ala8 with the NE2 atom of His42$^{SYM1}$, the OE1 and NE2 atoms of Gln9 with the carbonyl oxygen atoms of Ile31$^{SYM1}$ and Ile32$^{SYM1}$, and the O and N atoms of the main chain of Ala11 with the corresponding atoms of Gln34$^{SYM2}$, as well as by the salt bridge between the side chains of Arg12 and Asp87$^{SYM1}$ (Figure 5b). The N-terminus of chain B is involved in interactions with the following symmetry-related molecules: SYM3 (4_455) and SYM4 (4_555). In this case, the number of visible interactions is reduced to hydrogen bonds between the carbonyl oxygen of Pro13 with the OG atom of Ser45$^{SYM3}$ and the NE2 atom of Gln47$^{SYM3}$, and between the NE2 atom of Gln9 with the OD1 and OD2 atoms of Asp69$^{SYM4}$. In addition, several water molecules mediate the formation of non-direct hydrogen interactions between the N-terminal parts of chains A and B and symmetry-related protein molecules. Unfortunately, the quality of the calculated electron density maps decreased when Ala6 in chain A and Gln9 in chain B were reached, along with the observed perturbations due to solvent molecules, artifacts, and noncontinuous or weak density maps that cannot be reliably interpreted.

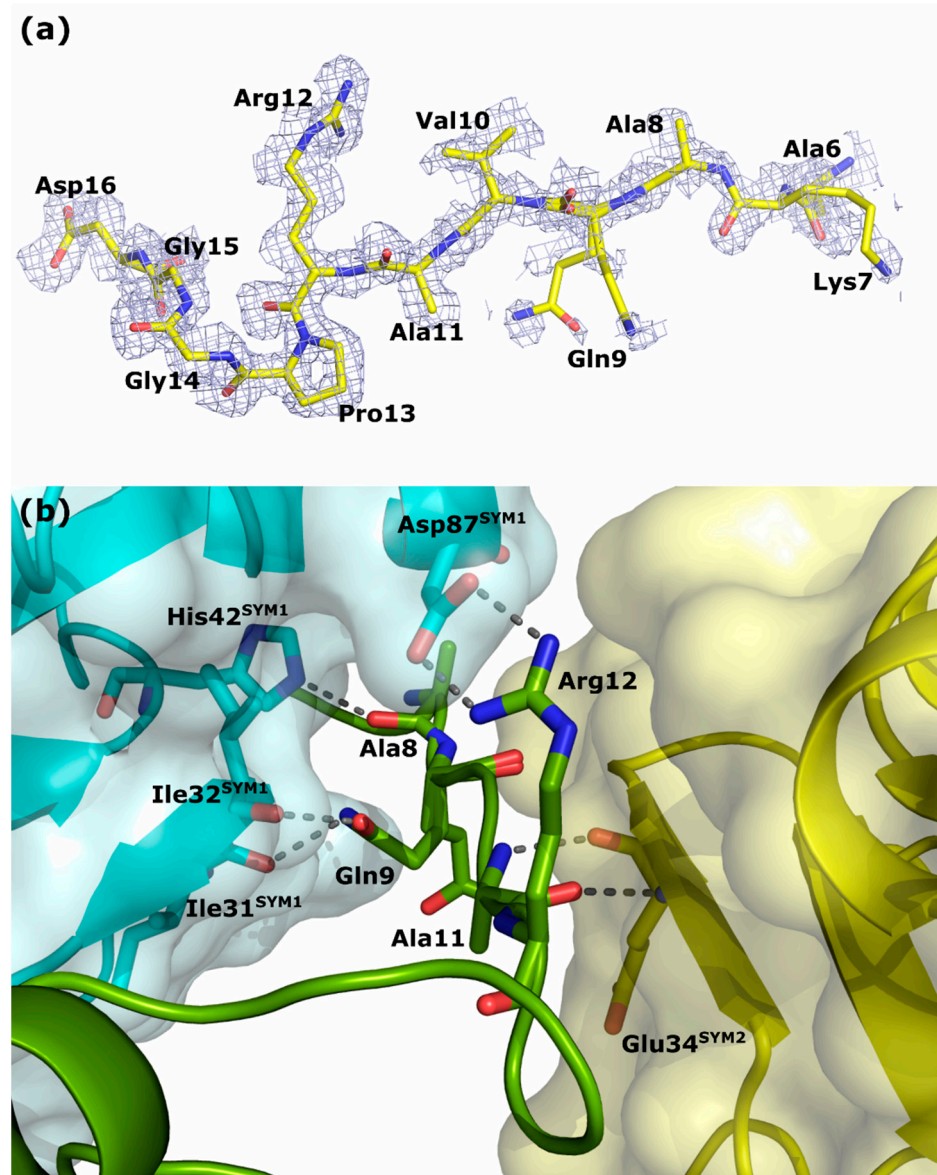

**Figure 5.** The N-terminal fragment of chain A of the hHINT1 protein. (**a**) The protein chain is shown in the stick representation. The electron density map 2 $F_O-F_C$ (colored light blue) at the 0.6σ level was derived from data at 1.43 Å resolution (PDB entry 6g9z). (**b**) Location of the N-terminus (green) between the symmetry-related molecules SYM1 (cyan) and SYM2 (yellow). Hydrogen bonds formed between amino acid residues of the N-terminal part of chain A and the symmetry-related molecules are shown as dashed black lines. Amino acid residues involved in the interactions are shown in stick mode. The red and blue sticks indicate oxygen and nitrogen atoms, respectively.

### 3.4. Analysis of the Nucleotide Binding Site/Ligand Molecule

HINT proteins bind purine nucleosides and nucleotides, although their nucleotide-binding motif differs in amino acid sequence from other nucleotide-binding proteins [51]. An analysis of the data for the hHINT1 form II (based on 2 $F_O-F_C$ and $F_O-F_C$ electron density maps) showed that no nucleotide ligands are bound in the cleft usually described as the active site [29,30], but we detected the presence of a single molecule of the L-malate ion in the binding pocket of chain B (Figure 6a). This ion was used as a component of the crystallization buffer, because the MMT buffer from the PACT++ crystallization screen consisted of DL-malic acid, MES and Tris base in the molar ratios 1:2:2, with a final buffer concentration of 100 mM. We observed that only the L-stereoisomer from the DL-malic acid mixture was recognized and bound by the HINT1 protein. L-malate interacts directly

with the protein chain via four hydrogen bonds formed with three protein side chains: Asn99 (ND2 atom of Asn99 and O2 atom of L-malate), His112 (NE2 atom of His112 and O2 and O4A atoms of L-malate), and His114 (NE2 atom of His114 and O2 atom of L-malate). Additional interactions are formed by water molecules with the backbone of residues Asn99, Ser107, Val108, and His112 (Figure 6b), and with the side chain of the Asp43 residue (not shown in Figure 6). Although the binding of this molecule is rather incidental than intentional, this, along with the use of a different type of buffer, may have contributed to the changes in crystal packing that led to the stabilization of the N-terminus. This information can also be used to design and verify the binding properties of other compounds being developed as inhibitors/substrate analogs for HINTs.

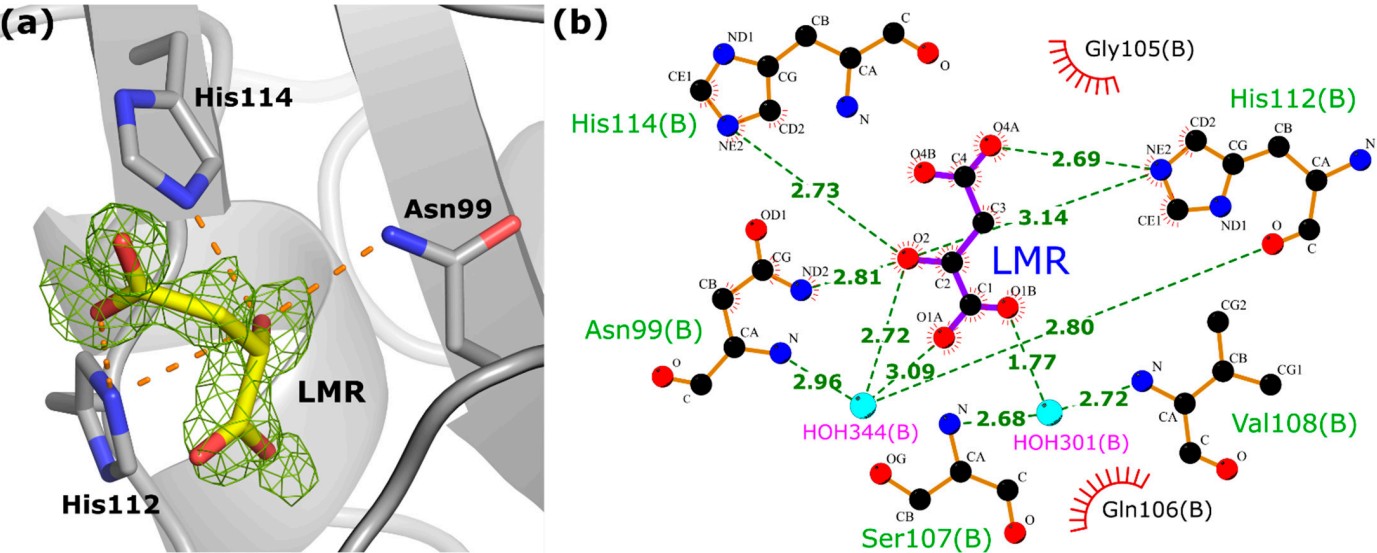

**Figure 6.** The nucleotide binding site of hHINT1 with bound L-malate anion. (**a**) The view of the binding site indicating Asn99, His112, and His114 residues involved in the L-malate ion direct binding. The hydrogen bonds between the ligand atoms and the nearby protein side chains are shown as dashed orange lines. The OMIT $F_O - F_C$ map of the L-malate molecule is contoured at the 3.0σ level based on data with a resolution of 1.43 Å (PDB entry 6g9z). (**b**) Schematic diagram of the binding site of the hHINT1/L-malate complex. Hydrogen -bonding interactions between L-malate atoms and nearby side chains and water molecules are shown as dashed lines. C atoms are shown in black, O atoms in red and N atoms in blue. Water molecules are shown in cyan. The L-malate ion is labeled as "LMR", according to the code used in PDB. This diagram was prepared with LIGPLOT [52].

## 4. Conclusions

The structure of a new crystal form of the human HINT1 protein (referred to here as form II), crystallized in space group $P2_1 2_1 2_1$ has been determined and refined to 1.43 Å resolution. It is characterized by a different crystal packing than those previously published. By changing the crystallization conditions, especially the type of buffer used, crystals with a higher degree of packing and a larger number of crystal contacts between symmetrically connected molecules could be obtained. The tighter crystal packing caused stabilization of the N-terminal fragment of the HINT1 polypeptide chain by forming new interactions with neighboring molecules. The stabilized fragment of the N-terminus was found using the obtained diffraction data. We also found that HINT1 selectively binds the L-malate ion from the mixture of stereoisomers in the crystallization buffer, which was not previously described.

**Author Contributions:** Overexpression and purification of the protein, A.M.; crystallization, X-ray data collection, structure solution and refinement, data visualization, validation, deposition and analysis, writing—original draft, R.D.; writing—review and editing, supervision, B.N. All authors have read and agreed to the published version of the manuscript.

**Funding:** This research was financially supported by CMMS PAS Statutory Funds, Poland.

**Data Availability Statement:** Atomic coordinates and structural factor amplitudes of the human HINT1 protein with visible extended N-terminus are available in the Protein Data Bank as entry 6g9z.

**Acknowledgments:** Diffraction data were collected at the BL14.1 beamline at the BESSY II electron storage ring at Helmholtz-Zentrum Berlin (HZB), Germany. We would especially like to thank Martin Gerlach for his help and support during the experiments. The authors thank Piotr Guga for constructive suggestions and discussions and Ewelina Wielgus for mass spectrometric analysis.

**Conflicts of Interest:** The authors declare no conflict of interest.

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
