# Peer review of "Crystal Packing Differences as a Key Factor for Stabilization of the N-Terminal Fragment of the Human HINT1 Protein"

_crystals, doi:10.3390/cryst13081197_

Round 1

Reviewer 1 Report

In this paper, authors reported the novel crystal structure of human HINT1 involved in nucleotide metabolism. This structure model contains a fragment of the N-terminal part that was not determined in the previously deposited structures. The crystals in this study have different crystal packing from the previously reported crystals. In addition, the authors found that L-malate bound to the nucleotide binding site of HINT1 in the crystal structure. Although there are some imperfections in this manuscript, they can be addressed and corrected to make it suitable for publication. I referred to the annotation reports and coordinates released in the Protein Data Bank (PDB ID: 6g9z) during my peer review.

Major comments:

Line89:

I couldn't find Table 1 in this manuscript.

Line 120:

While the manuscript states "There were no residues in improper Ramachandran conformation...", the PDB Annotation report indicates that Val10 in chain B has been detected as Ramachandran outlier. However, the absence of this sentence does not diminish the value of this paper.

Line 183:

There are fewer descriptions about the N-terminus of the chain B. Could you explain why the visible/modelled region was shorter in the N-terminus of Chain B?

Line 213 and Figure 6:

I felt a distinct discomfort with the hydrogen bonds shown in Figure 6. Using LIGPLOT+, I verified the protein-malate interaction, revealing the presence of hydrogen bonds solely between NE2 atom of His114 and O2 atom of L-malate, as well as between NE2 atom of His112 and O4A atom of L-malate, respectively.

Authors should discuss chemical bonds after verification based on stereochemistry.

Finally, I have a fundamental question. What is the reason why the crystals were obtained with the new packing in this work? The combination of PEG and neutral pH buffer is a common crystallization solution, but could that be the reason for the special packing? Or maybe the random L-malate that came in caused it? I think it would be a better paper if there was some discussion about the essential cause of the unique packing.

Minor comments:

Line 78:

Is this solution C4 instead of D4 in PACT++2?

Line79:

The term 'MMT buffer' is not widely known or commonly used. Could you provide more detailed information regarding its composition? Specifically, since malate in MMT buffer is an important additive molecule in this study, including the specific components and their concentrations in MMT buffer would greatly enhance the understanding of your paper.

Line 220:

Typo. “Figure 5” should be “Figure 6”.

Author Response

Answer for Reviewer 1

Answers for major comments:

  • Line 89: Table 1 has been added to the manuscript.
  • Line 120: We based our description on the PROCHECK report, in which Val10 of chain B was not flagged as a Ramachandran outlier (see attached fragment of the report below). On the other hand, the validation tool of COOT indicated two residues as outliers (Lys7 of chain A, Val10 of chain B). We assume that MOLPROBITY is used in the generation of the validation report. Therefore, we have updated this information in Table 1 and in the manuscript.
+- - - - - - - - - - <<< P R O C H E C K S U M M A R Y >>>- - - - - - - - - -+
|                                                                            |
| C:/data_cryst_new/hint1_apo_02/hint1_apo_02_ref_22.pdb   1.4  641 residues |
|                                                                            |
+| Ramachandran plot:   88.8% core   11.2% allow    0.0% gener    0.0% disall |
|                                                                            |
+| All Ramachandrans:    4 labelled residues (out of 235)                     |
| Chi1-chi2 plots:      0 labelled residues (out of 144)                     |
|                                                                            |
| Main-chain params:    6 better     0 inside      0 worse                   |
| Side-chain params:    5 better     0 inside      0 worse                   |
|                                                                            |
+| Residue properties: Max.deviation:    11.3              Bad contacts:    0 |
+|                     Bond len/angle:    5.0    Morris et al class:  1  2  2 |
+|     2 cis-peptides                                                         |
| G-factors           Dihedrals:  -0.23  Covalent:   0.02    Overall:  -0.08 |
|                                                                            |
| M/c bond lengths: 95.8% within limits   4.2% highlighted                   |
| M/c bond angles:  96.8% within limits   3.2% highlighted                   |
+| Planar groups:    90.1% within limits   9.9% highlighted                   |
|                                                                            |
 +----------------------------------------------------------------------------+
  • Line 183: We have chosen to describe in detail the fragment of the N-terminus of chain A because it is the longer visible fragment compared to chain B. One of the reasons for worse visibility of chain B was the poorer stabilization of the B chain, probably caused by the different environment of this fragment by symmetric molecules. The N-terminus of chain A is flanked by one symmetric A molecule and one symmetric B molecule, while the N-terminus of chain B is flanked by two symmetric B chain molecules.
  • Line 213/Figure 6: We analyzed hydrogen bonds between ligand and protein side chains using COOT. Compared to the results of LIGPLOT+ in standard mode, two hydrogen bonds were omitted (LMR O2 … Asn99 ND2 – 2.77Å and LMR O2 … His112 NE2 – 3.20Å), which should be added in PDB file used for calculation/image preparation. Figure 6 has been updated with correction of the description in the section “Analysis of nucleotide/binding site”. L-malate was selected based on the Fo-Fc electron density map, which D-malate could not match.
  • According to the last question in the major revision section, we think that the change of the buffer used together with the binding of one of its components was only one possible factor that could affect the unique crystal packing. Crystallization of HINT1 was performed many times in our laboratory with repeatable results, using different crystallization conditions, but mainly based on PEGs. For the first time, MMT buffer was used, resulting in a different crystal packing. Perhaps it was just a coincidence, but we believe it is still worth publishing this result. Additional discussion on this has been included under “Analysis of nucleotide binding site/ligand molecule” and “Conclusions”.

Answers for minor comments:

  • Line 78: We agree with the reviewer's suggestion. The description of the crystallization conditions of PACT ++ 2 has been changed from C4 to D4. We used an old description that was used earlier when the 24-plate format was common. We noticed that the manufacturer changed the descriptions, so we updated it to the current description.
  • Line 79: Short information about the MMT buffer composition has been added to the section “Analysis of nucleotide binding site/ligand molecule”.
  • Line 220: We changed number of the indicated drawing from “Figure 5” to “Figure 6”.

Reviewer 2 Report

The manuscript “ Crystal packing differences as a key factor for stabilization of the N-terminal fragment of the human HINT1 protein” by Rafał Dolot et al. describes the crystal structure of an alleged form II of HINT1 (crystallizing in the SG19, P212121). The manuscript is generally well structured and prepared and features the crystallization and refinement of the HINT6g9z structure along with MS, for confirming the molecular weight/sequence  of the protein. The 6g9z has been deposited and is available in the PDB. Both N and C ends in the Structure Validation Report are flagged in red and yellow, thus acknowledging  the movement and somewhat lesser  Fo electron density. I have  only one  concern that  is related ti the elevated R/Rfree factors. Usually the difference between the R/Rfree is expected to be  up to 5% and  in addition statistically the Rfactors -resolution is  +5 so  for  1.4+ 0.5 the R  factors should be  around  20%. Have you tried to refine  the  structure  without the dangling C/N tails?

Although a table with the   Data and refinement statistics is  present in the  Validation Report  It would be nice to have it in the  text.

I believe you used the D4  condition of the JBS PACT ++ 2. In the  paper of  Kimberly M Maize et al. (5klz)  other SG are reported  e.g. tetragonal, monoclinic … A rapid  superposition  of the two structures  does not suggest striking structural  deviation, just packing  differences. I would recommend to add a few more lines of comparisons between the different SGs.

Author Response

Answers for comments:

  • We agree that R/Rfree should be lower for the obtained resolution data, but it is still within acceptable limits. The C-terminal parts of chains A and B are really well aligned and stabilized, also the B-factors of these parts are at low B-factor level. We reviewed the reviewer's suggestion and performed additional refinement for the structure truncated to Arg12 for both chains, but R/Rfree increased to 24.38/29.00, indicating that our extended model is well fitted.
  • Table 1 is attached to the manuscript. We apologize profusely for the absence of this table, which was inadvertently deleted during the last stage of manuscript preparation
  • The description of the crystallization conditions of PACT ++ 2 was changed from C4 to D4. We used an old description that was used earlier when the 24-plate format was common. We found that the manufacturer changed the descriptions, so we updated it to the current description.
  • We have added an additional discussion of crystal packing observed on various SGs for HINT1.

Reviewer 3 Report

The authors present a structure of human HINT1 protein determined by X-ray crystallography. The methodology and results are sound, and the figures are well presented. 

The novel findings of this work arise from the different crystal packing that was observed compared to previous structures. The N-terminal region of the  HINT1 protein is disordered, and therefore not resolved in prior structures (most published structures have 11 N-terminal residues unmodelled). In this work, the N-terminal region is shown to interact with a neighbouring molecule via crystal packing interactions, therefore an additional six residues have been modelled (AKAQVA). Five N-terminal residues remain disordered and unmodelled.

For data availability, I'd like to thank the authors for releasing the coordinates of the structure prior to publication (PDB accession code 6g9z).

Line 216: Please change the word "random" to "incidental". The usage of "random" in this context is rather informal or colloquial.

I suggest including a statement in the discussion to highlight that the ordered state observed here is an artefact of crystal packing, and is therefore not likely to be physiologically relevant. In solution, the N-terminal region is likely to remain disordered unless there are stabilising protein-protein interactions. 

Optionally, if there is a valid reason, the authors may wish to comment why the Rfree value is high (0.277), especially for comparable structures of similar resolution. 

Author Response

Answer for comments:

  • Line 216: The word "random" was changed to "incidental", according to Reviewer's suggestion;
  • Based on the next suggestion, a fragment about the observed state was added to the manuscript in the section "Structure of the N-terminus";
  • We agree that R/Rfree should be lower for the obtained resolution data, but it is still within acceptable limits. To test the effect of the most dangling N-terminal parts (with the highest B-factors) on the refinement indicators, we performed an additional round of calculations for the structure truncated on Arg12 for both chains, but R/Rfree increased to 24.38/29.00, indicating that our extended model is well fitted.

Round 2

Reviewer 1 Report

I appreciate your thorough and precise response to the revisions.
I recommend accepting your paper for publication.